# Polarization and trust in the evolution of vaccine discourse on Twitter during COVID-19

Ignacio Ojea Quintana[1]*, Ritsaart Reimann[2], Marc Cheong[3], Mark Alfano[2], Colin Klein[1]

1 School of Philosophy, The Australian National University, Canberra, ACT, Australia, 2 Department of Philosophy, Macquarie University, Sydney, NSW, Australia, 3 Centre for AI and Digital Ethics, Faculty of Engineering and IT, University of Melbourne, Melbourne, VIC, Australia

* ignacio.ojea@anu.edu.au

**Data Availability Statement:** Data are available from Open Science Framework: https://osf.io/b65uc/.

**Funding:** This paper was supported by Australian Research Council Grant DP190101507 (to Colin

## Abstract

Trust in vaccination is eroding, and attitudes about vaccination have become more polarized. This is an observational study of Twitter analyzing the impact that COVID-19 had on vaccine discourse. We identify the actors, the language they use, how their language changed, and what can explain this change. First, we find that authors cluster into several large, interpretable groups, and that the discourse was greatly affected by American partisan politics. Over the course of our study, both Republicans and Democrats entered the vaccine conversation in large numbers, forming coalitions with Antivaxxers and public health organizations, respectively. After the pandemic was officially declared, the interactions between these groups increased. Second, we show that the moral and non-moral language used by the various communities converged in interesting and informative ways. Finally, vector autoregression analysis indicates that differential responses to public health measures are likely part of what drove this convergence. Taken together, our results suggest that polarization around vaccination discourse in the context of COVID-19 was ultimately driven by a trust-first dynamic of political engagement.

## Introduction

Trust in vaccination remains high, but is eroding in many parts of the world [1, 2]. Decreased confidence in the safety, efficacy, and importance of vaccination may manifest as open skepticism and conspiracy theorizing. It may also manifest more subtly in vaccine hesitancy, which leads to questioning the need for vaccination. Hesitancy comes in degrees: some people are 'accepters' while others are 'fence-sitters' or 'rejecters' [3]. Confidence is impacted by lack of information and access to misinformation, and by distrust of medical and government sources [4–6].

When people lose trust in medical experts and public health officials, they tend to turn to other sources, including social media. Social media sites optimize for engagement, rather than other measures such as information veracity or epistemic well-being [7–9]. This is concerning because exposure to negative opinions about vaccines on social media has been shown to be

Klein). The funders had no role in study design, data collection and analysis, decision to publish, or preparation of the manuscript.

**Competing interests:** The authors have declared that no competing interests exist.

among the strongest predictors both of expressing such opinions oneself [10] and of failure to vaccinate [11].

Expressions of vaccine hesitancy on social media, and in particular on Twitter, have been shown to co-vary with offline expressions of the same sentiment [12]. Furthermore, a pair of recent, pre-COVID-19 studies found that the English-language discourse about vaccines on Twitter is highly polarized, that the anti-vaccine camp has greater reach and receptivity, and that discussants tend to rely on and amplify just a few, non-independent sources [13, 14].

Most of the pre-pandemic discourse about vaccination revolved around well-established immunizations to well-understood childhood diseases, seasonal influenza, and human papillomavirus. Well-established vaccines suffer from the 'curse of success' because their widespread administration has in many cases reduced incidence levels of the relevant diseases to near-zero. By contrast, in the early part of 2020 COVID-19 was not well understood, and vaccines against it were months away. Debates about vaccine efficacy and side-effects were therefore conducted in the absence of empirical evidence.

To shed light on the evolution of social media discourse around vaccines in the first few months of the pandemic, we conducted an observational study of Twitter discourse around vaccine topics in general between December 27th 2019, and May 5th 2020. This time frame spans 75 days prior to the World Health Organization's March 11th 2020 pandemic declaration through to 75 days after. The WHO's declaration was not merely symbolic; it also triggered a series of institutional responses, some of which (we show) significantly affected vaccine discourse.

Previous work on Reddit has demonstrated the utility of mixed methods for understanding the evolution of online engagement in complex domains [15]. We set out to use a combination of methods to answer the following three research questions:

**RQ1**. Which groups are most important in the English-language discourse around vaccines on Twitter?

**RQ2**. How did vaccination-related engagement and discourse change over the first five months (12/2019–05/2020) of the pandemic?

**RQ3**. What social forces might help in explaining observed changes in engagement?

As explained in the coming sections, we used modularity clustering to answer the first question, Linguistic Inquiry and Word Count (LIWC) to answer the second, and vector autoregression analysis (VAR) for the third. We found that authors cluster into five interpretable groups, and that the discourse was greatly affected by American partisan politics. Since we did not limit data collection to the U.S. nor any other territory, the prevalence of American politics within our analysis is partly due to our focus on English-language discourse; partly due to the fact that the vast majority of Twitters English speaking user-base is located in the United States; and partly a reflection of the extent to which American political cleavages define global online discussions, at least when those discussions are carried out in English. Important to note therefore is that even though we distinguish 'Democratic' and 'Republican' clusters of users, it is not the case that all users in our data set are based in the United States. This is especially clear with respect to the coalitions that emerge between 'Democrats' and 'Public Health' on the one hand and 'Republicans' and 'Antivaxxers' on the other, for neither health institutions nor Antivaxxers are unique to U.S. discourse.

Finally, since 'Antivaxxer' is a loaded and much contested concept, it bears spelling out that our use of the term is descriptive rather than normative and, more than anything else, reflects our interpretation of the top accounts (e.g., *StopVaxTyranny*) and most popular hashtags (#VaccineRoullete) that *connect* members of this cluster. Concomitantly, we operationalize

this label simply as *a set of social relations around anti-vaccination attitudes*. This definition is deliberately open-ended, for tracking how these attitudes and social relations developed over the course of the pandemic is a core objective of the current paper.

With this in mind, our linguistic analysis zeros in on both the moral and non-moral language used by the various communities. In so doing, we identify significant patterns of thematic converge and divergence within and between both sets of coalitions as the pandemic unfolds. Finally, we find that these patterns can be partially explained at hand of VAR analysis, which shows that different groups responded differently to public health interventions. This result, combined with the observed changes in language use, leads us to the conclusion that polarization in the context of COVID-19 can be explained by a trust-first dynamic of political engagement: the realignment of interests over the course of the pandemic was driven less by shared information or shared values, and more by a proclivity of individuals to trust those with whom they already had some previous pattern of interaction.

The next section describes the methodology with respect to data collection, network construction, community detection, linguistic study, and time series analysis. The section after presents the results as organised by the three guiding research questions. The final two sections include a discussion of the results, statement of limitations, and a conclusion with some policy suggestions.

## Materials and methods

### Data collection

We queried the Twitter Streaming API with a series of vaccination-related keywords, hashtags, and short expressions between December 2019 and June 2020. Some examples include: 'vax', 'vaxxed', 'vaccine', 'vaccination', 'antivax', 'anti-vax', 'anti vax', '#vaxsafety', '#vaccineswork', '#novax', '#antivax'. The choice of words was done following similar literature on vaccination discourse on Twitter [16], and with the goal of trying to capture a wide spectrum vaccine related attitudes. See S1 Appendix for a complete list. Taking as a reference point the date the World Health Organization declared COVID-19 a pandemic (March 11th 2020), we divided our data set into a symmetric time-span of tweets comprised of approximately 1.3 million original tweets and 18 million retweets between December 27th 2019 and May 26th 2020.

Because we were interested in the interactions between users and groups, we focused on *retweets* rather than tweets with original content. The phenomenon of 'signal-boosting' members of one's own group is familiar to anyone who has spent time on Twitter, where people tend to retweet messages published by those they view as co-partisans or allies [17]. Retweets generally signal endorsement of content and an attempt to signal-boost. More specifically, retweets serve three purposes: to spread tweets, to start a conversation, and to draw attention to the originating user [18, 19]. Retweets thus play an important community-building role.

A tweet is either wholly original content, a quote tweet (which retweets and adds commentary), or a retweet of either an original or quote tweet. We considered retweets of both original and quote tweets in our analyses. We also examined retweeted content. If an original tweet was retweeted, we considered the content and author to be that of the original tweet. If a quote tweet (or a series of quote tweets) was retweeted, then we considered the retweeted content and author to be that of the most recent comment. This preserves the endorsement flavor of retweets: if $x$ says something, $y$ quotes to disagree, and $z$ retweets $y$'s disagreement, it is likely that $z$ also disagrees with $x$. Note that the Twitter API functions in such a way that intermediate retweets are not stored: if $y$ retweeted $x$'s tweet $T$, and $z$ retweets $y$'s retweet, the data will show only a retweet of $x$ by $z$ (omitting $y$).

## Network construction and community detection

We generated a *retweet network*, a weighted directed network where nodes are authors and the weight of an edge from node *u* to node *v* represents the number of times that user *v* retweeted user *u*. Self-retweeting was discarded. Users that only retweeted but never authored an original tweet were discarded. Retweet networks have been used before to study community engagement and the spread of fake news [13, 14, 20]. We considered only the principal weakly connected component of the network. The full network has $\sim$ 380K nodes and $\sim$ 3.6M edges. To test for biases in our data, we did a power law analysis and found that the network follows a statistically significant power law distribution (see S1 Appendix for details).

Using time-lag analysis for users we identified around 500 bots in our data set (see S1 Appendix for details). We did not remove the bots for two reasons. First, the overall small number of bots (500 over $\sim$ 380K total users identified) gives us confidence that the linguistic results will not be seriously skewed by any oddity in bot content. Second, we note that we have treated bots and non-bot users uniformly, in the sense that bots are themselves only included if they make original tweets as well. These are likely to be unusual bots (or, perhaps, hybrid human/bot accounts), and the fact that they both retweet and post original tweets is reason to believe that they would play the same functional role in bringing together communities as would a similarly situated human user.

Modularity optimization is an unsupervised method used for community detection. The modularity of a network measures the strength with which a network can be divided into groups. The measure works by computing the fraction of edges that fall within a given community minus the expected fraction if edges were distributed at random, but keeping the same (weighted) degree distribution. Therefore, a high modularity indicates that members of a community are unexpectedly bound to each other, holding node centrality fixed and having randomness as a baseline.

The most well studied and standard algorithm for modularity maximization is Louvain, developed by [21]. We used Gephi's implementation of it because it can handle weighted and directed networks like our retweet network, and allows for different resolutions as developed by [22]. We ran the community detection algorithm including randomization and edge weight. Furthermore, we repeated the implementation multiple times and with different resolution values and results reported in the next section remained consistent.

To characterize the communities we considered the top verified accounts in each cluster, as well as the typical hashtags used by the communities. To confirm that the communities extracted by our network modularity analysis were also thematically unified, we ensured that standard machine learning classifiers, using a variety of approaches, could classify users on content at a level well above chance (see S1 Appendix for detailed results). We then restricted further analyses to users in the top 5 identified communities.

## Corpus-based analysis of retweets

To study the corpora of tweets per group, we preprocessed the data to exclude non-English words, characters and symbols, as well as English stop-words. The analysis employed a frequency-based approach modeled on Linguistic Inquiry and Word Count (LIWC) [23, 24], which has proven useful in other recent analyses of COVID-related social media discourse [25, 26] In the interest of open science, we used the R package *LIWCalike* [27], which imitates and expands the functionality of LIWC. As is standard in LIWC analyses, we did not perform any stemming/lemmatizing of the corpora.

An advantage LIWC is the ability to create and share custom dictionaries for categories of interest. Recent interdisciplinary work has shown that it is possible to extract a moral signal

from natural language using various tools [28]. For this analysis, we decided to use the custom Moral Foundations Dictionaries (MFD), which are keyed to various moral concerns. Details of these dictionaries are available here (MFD).

The moral foundations dictionaries measure the number of words in a text associated with care (versus harm), fairness (versus unfairness), authority (versus insubordination), loyalty (versus disloyalty), and sanctity (versus corruption). These domains or 'foundations' feature in Moral Foundations Theory [29] and are conceived of as topics towards which individuals are differentially sensitive. MFD includes two sub-dictionaries (one related to virtues, the other to vices) for each foundation.

## Time series analysis

To further examine the relationship between groups and tweets over the course of the studied period, we performed a vector autoregression (VAR) analysis of retweets. VAR is an extension of multiple regression that attempts to fit the value of variables at time $t$ using their value at time $t - l$, where $l$ is a chosen lag. VAR is widely used in econometrics [30] and has been used in the study of online time series data to, e.g., investigate the relationship between mass shootings and online interest in gun control and gun purchasing [31]. We used the Python package *statsmodels* (v0.12.2) for all VAR-related analyses [32].

For the endogenous variable we used retweets of tweets by a particular cluster indexed to the day of retweeting, which is a measure of the influence of each cluster. We chose an *a priori* lag of 1 day, as the influence of tweets tends to fade quickly. Unlike (e.g.) raw tweet counts or retweet activity—both of which have obvious trend increases across our time series—influence is stationary in each group (AD Fuller test, $p \leq 0.05$ uncorrected).

We examined VAR coefficients at lag 1 for the influence of each of the five groups. Using the generated model, we also performed tests for Granger causality between groups. To examine the effect of public health interventions on the relationship between groups, we re-ran our analyses using a publicly available data set of health measures [33], using the aggregated US response as our model.

## Results

### RQ1: Community characterization and classification

**Community characterization.** We obtained a network modularity score of 0.608 for our retweet network. The implementation found 231 total communities, with the top five largest communities comprising ∼80% of the population and the top eight largest comprising ∼95%. Manual inspection showed that similar community partitions emerged in repeated implementations of the algorithm, and with different resolution values.

We focused on the top five communities. These not only contain ∼ 80% of nodes, but are responsible for ∼ 90% of retweets. Communities beyond the top five also tended to cluster around non-English-language accounts, which limits the utility of our dictionary-based tools. For all further analyses, we considered the subgraph of our full graph containing members of these five groups, and only content originated and retweeted by those members.

Using representative nodes and popular hashtags for each group, we were able to interpret each cluster and add descriptive labels. Note that in adding these labels, we neither attempt nor purport to offer rigid definitions. Instead, we pick out broad patterns within and between communities that can help orient our thinking and guide the ensuing discussion. Worth reiterating here is that even though we did not limit data collection to the United States, the vast majority of users are U.S. based. With this in mind, the 'Democratic' cluster (Blue) includes Democratic politicians, center-left media, and self-identified Democratic partisans. The

'Republican' group (Red) similarly contains many Republican politicians (including then-President Donald Trump), self-identified Republican partisans, and right-leaning media outlets, at least some of which contributed to the spread of COVID-related misinformation [34, 35]. 'Public Health' (Yellow) is largely made up of independent health professional and public health institutions, including the World Health Organization, the Centre for Disease Control and Prevention, and various European health agencies. This cluster is most clearly distinguished from the 'Antivax' community (Black), which features traditional anti-vaccination accounts such as *StopVaxTyranny* and promotes hashtags like #VaccineRoullete. Note again that we use this label in a descriptive rather than normative sense. Note as well that even though most of the accounts in this cluster are U.S. based, it also features users from Europe and Africa. Finally, and in addition to being the only community that is not dominated by U.S. accounts, the 'Unorthodox' group (Green) is considerably more heterogeneous than the previous four; we return to characterizing it further below.

As Fig 1 shows, the retweet network was highly polarized before and after the declaration, and features two distinct alliances. At one pole, we see that Democrats and Public Health organizations are already closely connected and that these connections increase as time passes; likewise, at the other pole, Antivaxxers and Republicans appear to interact a lot initially and even more so after the pandemic declaration. This topography is consonant with various recent studies showing that attitudes toward vaccination and other preventive measures aimed at reducing the spread of COVID-19 are strongly sorted along partisan lines [36–38]. In fact, political polarization has been found to be the primary driver of opposition to public health interventions [39–41], partially explaining the alliance between Antivaxxers and right-leaning users. Finally, we see that the Unorthodox community sits somewhere between these two poles, but is pulled slightly more toward the Antivax-Republican coalition. Fig 1 also shows a significant increase in engagement with vaccine discourse after the declaration; we discuss this in detail later in Table 2.

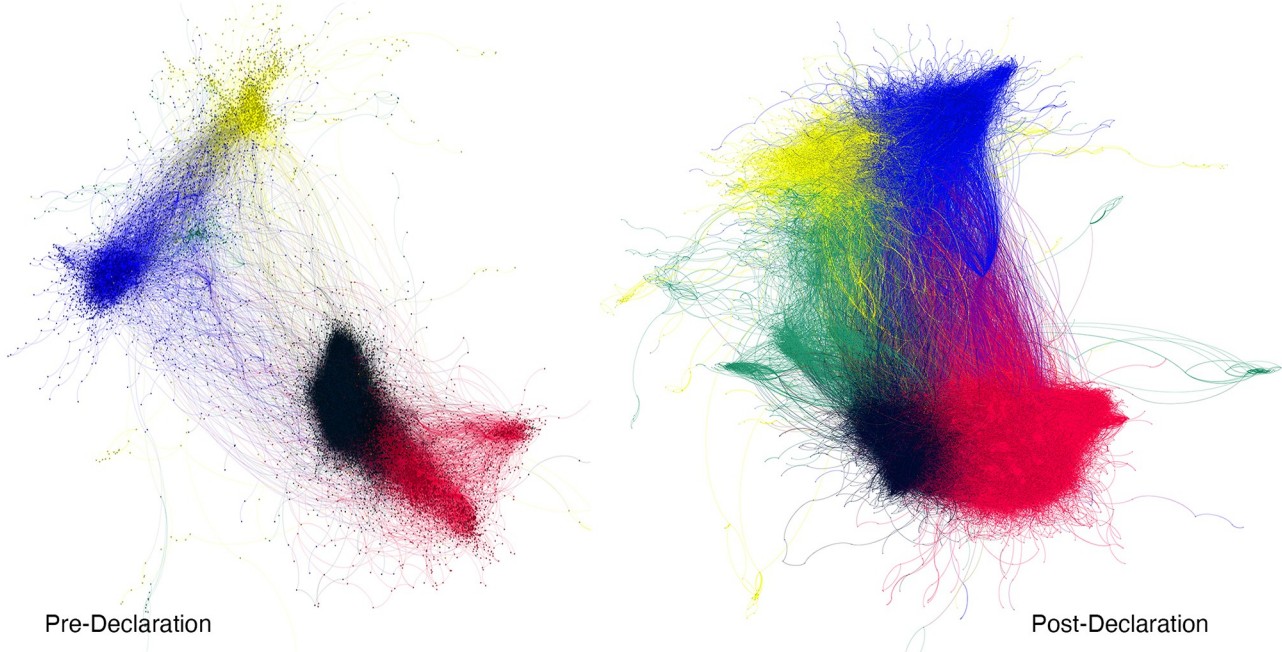

Pre-Declaration · Post-Declaration

**Fig 1. Visualisation of the retweet networks before and after the WHO declaration, color-coded by community.**

**Table 1. Summary statistics for the top five communities.**

| Community name | % of nodes | % of retweets | % of verified users | Popular Hashtags | Representative Nodes |
|---|---|---|---|---|---|
| Democrats | ∼ 24% | ∼ 20% | ∼ 10% | #moronpresident, #trumpslump, #gopvirus, #trumpgenocideforprofit, #trumpburialpits | JoeBiden, KamalaHarris, SenWarren, BillGates, CNN, nytimes, washingtonpost, ABC, businessinsider, MSNBC, guardian (theguardian), TIME, BBCWorld |
| Republicans | ∼ 18% | ∼ 35% | ∼ 2% | #kungflu, #notest, #boycottchina, #trumpCOVIDgate, #illuminati | realDonaldTrump, mikepence, RealCandaceO, WhiteHouse, FoxNews |
| Unorthodox | ∼ 16% | ∼ 6% | ∼ 3% | #AfricansAreNotLabRats, #AfricansAreNotGuineaPigs, #listentotheexperts, #locksouthafricadown, #coronavirusghana | BernieSanders, Trevornoah, spectatorindex, jacobinmag, NaomiAKlein, BBCAfrica, DrTedros, News24 (African Media) |
| Public Health | ∼ 13% | ∼ 7% | ∼ 9% | #epidemic, #hepatitisa, #immunoonc, #whatwedoinpharmacy, #scteenvax | CDCgov, WHO, EU_Health, UniofOxford, UNICEF, ProfPCDoherty (Nobel Laureate Immunology), VaccinesToday, CDCFlu, newscientist, CEPIvaccines, gavi |
| Antivaxxers | ∼ 8% | ∼ 22% | ∼ 0.8% | #illuminati, #praybig, #notest, #mykidsmychoice, #vaccineroulette | stopvaccinating*, StopVaxTyranny*, EpigeneticWhisp*, vaxxplained*, JustSayNo2Vax*, va_shiva, Jimcorrsays |

Table notes: To protect user privacy, only public figures (with a 'verified account' badge) and suspended/deleted accounts are listed as examples within. (i) *% of retweets*: These account for both retweeting and being retweeted within the network. (ii) *Popular Hashtags*: All of these are within the top 15 hashtags that each community used, after preprocessing and using a term frequency-inverse document frequency (*tf-idf*). (iii) *Representative Nodes*: Users marked with asterisks (*) have either been suspended or deleted by Twitter at time of writing. The typical reason for suspension is violation of the Twitter Terms of Service.

To get a better sense of these communities, Table 1 provides summary statistics for each group. Note that the contribution of each group dissociates somewhat from its size. Despite accounting for only ∼ 8% of nodes in the network, Antivaxxers contributed ∼ 22% of retweets. Similarly, Republicans made up just ∼ 18% of the network but were responsible for ∼ 35% of retweet activity. These results are consistent with findings by [13, 14] and suggest that these groups consist of extremely active and vocal individuals.

The first four clearly definable groups are consistent with extant research on automated community detection and characterisation on US-specific tweets during the dawning of the COVID-19 pandemic, circa early 2020 [42]. However, since our data-set covers a longer time frame, and we did not filter non-US tweets, we were able to identity a fifth community we called the 'Unorthodox' group. In addition to featuring accounts as diverse as those of Bernie Sanders, Trevor Noah, and BBC Africa; its top hashtags reflect both pro- and anti-vaccination attitudes. To make sense of these mixed impressions, we conducted a more careful inspection of this group's posts around day 100 of our data set, during which time their activity spiked. We found that this community contains many Africa-based users who became active in response to the suggestion that COVID-19 vaccines should be trialled in Africa [43].

For all groups, the vast majority of accounts were created before December 2019 [Democrats: 96%, Republicans: 88%, Unorthodox: 95%, Health: 96%, Antivaxxers: 86%]. If we compute engagement time of an author as the distance in days between their first and last tweet in our data set, Democrats were active for an average of 16 days, Republicans 17, Unorthodox 8, Public Health 18, and Antivaxxers 23. As one might expect, Antivaxxers and Public Health institutions engaged for a longer time span than the other communities, since they were often tweeting about vaccination issues before the pandemic started. It is also noticeable in Fig 2 that by the end, the conversation is dominated by Democrats and Republicans, hinting at how politicised the issue became (see also [34, 38]).

In addition to differences between groups in total numbers of retweets, there are also substantial variations in who each group retweets. Table 2 below shows the change in absolute

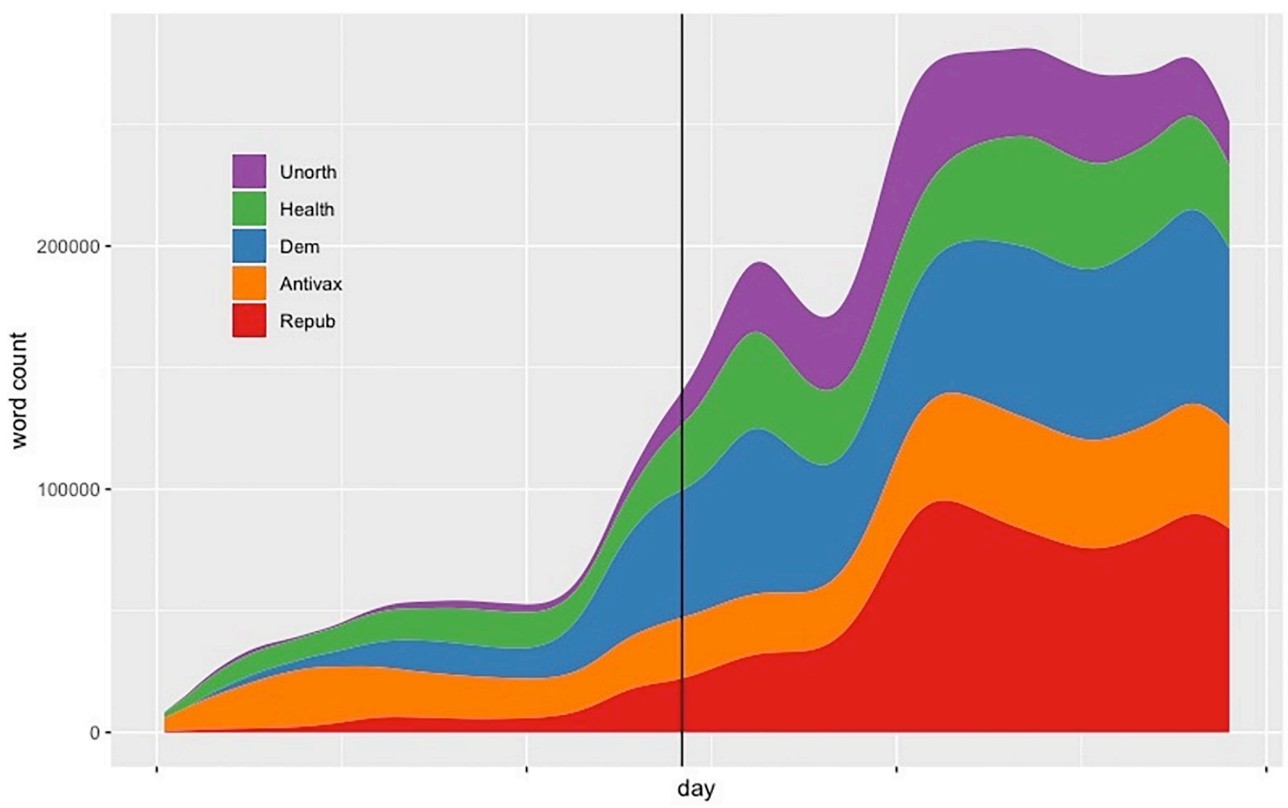

**Fig 2. Daily word count by community.** WHO pandemic declaration at the center.

numbers and ratios of post- to pre-declaration retweets. Unsurprisingly, the largest increases were in groups signal-boosting their own members. This depended in part on how active the groups were pre-pandemic: Antivaxxers and Public Health increased at the lowest rate, while Republicans retweeted Republicans at a vastly higher rate. There were also notable cross-group interactions, particularly between Antivaxxers, Republicans, and the Unorthodox community. In addition to frequently boosting each other's signal, both Republicans and Antivaxxers retweeted the Unorthodox more than that the Unorthodox retweeted either of these two groups. One possible explanation for this derives from our earlier observation that the Unorthodox community was specifically concerned with vaccine trials in Africa. Hence, despite expressing some hesitancy, it stands to reason that they only partially endorsed the general scepticism expressed by Antivaxxers and Republicans. While it is difficult to assess the

**Table 2. Changes in absolute numbers of retweets, expressed in 1000s of tweets, with ratio in parentheses.**

| ↓ \ → | Democrats | Republicans | Unorthodox | Public Health | Antivaxxers |
|---|---|---|---|---|---|
| Democrats | 452 (3.4) | 11 (3.1) | 20 (4.1) | 29 (3.7) | 5 (3.4) |
| Republicans | 7 (4.5) | 1151 (8.8) | 7 (9.2) | 2 (5.5) | 129 (9.6) |
| Unorthodox | 11 (2.4) | 4 (6.1) | 142 (5.2) | 3 (2.2) | 7 (7.3) |
| Public Health | 18 (2.6) | 2 (3.0) | 4 (3.7) | 108 (2.4) | 2 (2.7) |
| Antivaxxers | 2 (3.3) | 110 (4.8) | 8 (9.8) | 3 (2.9) | 101 (1.3) |

Table notes: Rows (↓) correspond to the retweeting community, while columns (→) correspond to the retweeted community.

authenticity of Republicans' and Antivaxxers' support for the Unorthodox, this asymmetry suggests that Antivaxxers and Republicans were more willing to promote the Unorthodox community's concerns than vice-versa.

**WHO pandemic declaration as a threshold.** We constructed our data set to be symmetric around the WHO pandemic declaration on March 11th 2020. To further justify this choice, we note that daily word count shows increased engagement across all communities over the course of the pandemic, and begins to spike around the declaration (Fig 2). In addition, we can see that Democrats, Republicans, and the Unorthodox begin to account for a larger share of the conversation, post-declaration. This is suggestive of a significant shift in discourse dynamics.

We also used time series methods to look for a structural break in the data (see S1 Appendix for details). We found evidence of a significant structural break about 5 days after the declaration of the pandemic. Part of this difference might have been due to a lagged response of tweets to the event. However, a more plausible candidate for the cause of the break is the ramp-up of public health measures in response to the pandemic itself.

## RQ2: The evolution of vaccine discourse

Our second research question concerned the evolution of vaccine discourse over time. We were interested in whether the moral and non-moral language used by the various communities reflects the patterns of polarization and alliance formation identified in the retweet network.

**Linguistic evolution.** There were 10 total corpora (5 communities × 2 periods, i.e., pre- and post-declaration). The corpus associated with each group more than doubled in size from pre- to post-pandemic declaration. The largest increase was among Republicans, whose corpus went from 533,620 words to 4,509,393 words, suggesting a surge in interest in a topic that previously had been of relatively little concern to these users.

We use hierarchical clustering to show similarities between the language used by groups before and after the pandemic declaration. For this part of our analysis, each community was associated with a vector corresponding to the *tf-idf* [44] score for each word.

As Fig 3 suggests, before the pandemic declaration, there were two thematically-unified discourses: one about politics, carried out by Republicans and Democrats, and another about health, carried out by Public Health and Antivaxxers. This doesn't mean that Democrats and

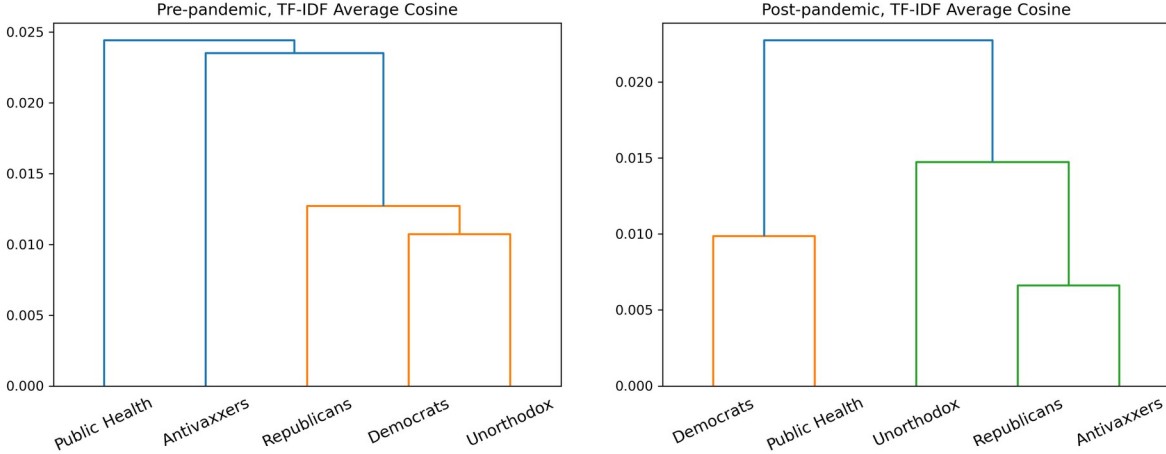

**Fig 3. Dendrogram showing hierarchical clustering for pre- and post-declaration tf-idf vectors.**

Republicans agreed: rather, they were talking about the same issues in a broadly similar way. Likewise, Public Health and Antivaxxers debated using much the same language.

After the pandemic declaration, however, we observe a reshuffling of the discourses: Democrats and Public Health start using the same language, while Republicans, Antivaxxers, and to some extent the Unorthodox become more linguistically similar to one another as well.

**Standard LIWC dictionaries.** Using standard LIWC dictionaries, we can shed further light on exactly how the language of these different communities shifted.

The language-based dendrograms in Fig 4 show that patterns in both moral and non-moral discourse evolved to match the patterns of social connection. Initially, the moral and non-moral language used by Antivaxxers and Public Health were most similar to one another, and the moral and non-moral language used by Republicans, Democrats, and Unorthodox were most similar to one another. However, after the pandemic declaration we find Republicans and Antivaxxers expressing the same moral and non-moral concerns, Democrats and Public Health expressing the same moral and non-moral concerns, and the Unorthodox again playing an ambivalent role in between the two polarities.

At a more granular level, Fig 5 shows 15 LIWC components that changed the most from pre- to post-declaration, shedding light on how the discourse evolved during the first few months of the pandemic. Across the board, we see decreases in 'Female', 'Family', 'Risk', 'Sexual', and 'Health'. The first three might be explained by a shift from a vaccine discourse traditionally centred around parental vaccination of children and the perceived risk thereof to one that focuses on the broader context of vaccination. The decrease in 'Sexual' is likely due to a comparative drop-off in discussions around human papillomavirus vaccination in particular. The drop-off in discussion of 'Health' is somewhat surprising; one possible explanation is that increased polarization means that the issue of vaccination was increasingly framed in political terms.

In contrast to these convergent trends, Democrats and Public Health score much lower than Antivaxxers and Republicans on the 'Anger', 'Body', and 'Feel' dictionaries post-declaration; suggesting a shift towards a more neutral, dispassionate mode of discussion. Perhaps more strikingly, we see a dramatic increase in discussions surrounding 'Home', 'Money', and 'Masculinity', especially among Republicans and Antivaxxers. The increase in 'Money' is likely related to rising unemployment, increasing economic uncertainty, and the NYSE's March 18th decision to close Wall Street [45, 46]. With respect to 'Home', we note that California

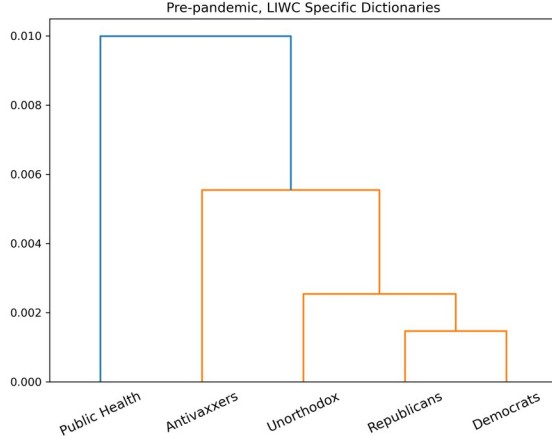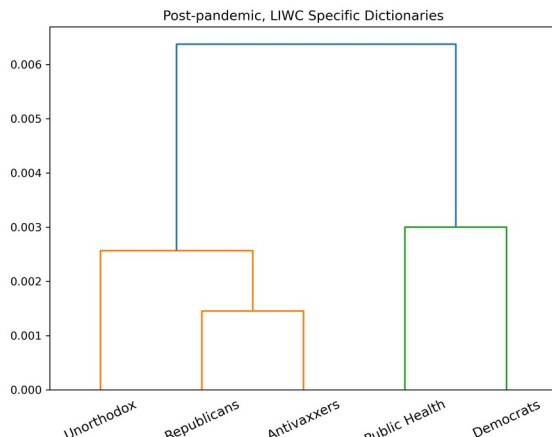

**Fig 4. Dendrogram for LIWC-Specific content.**

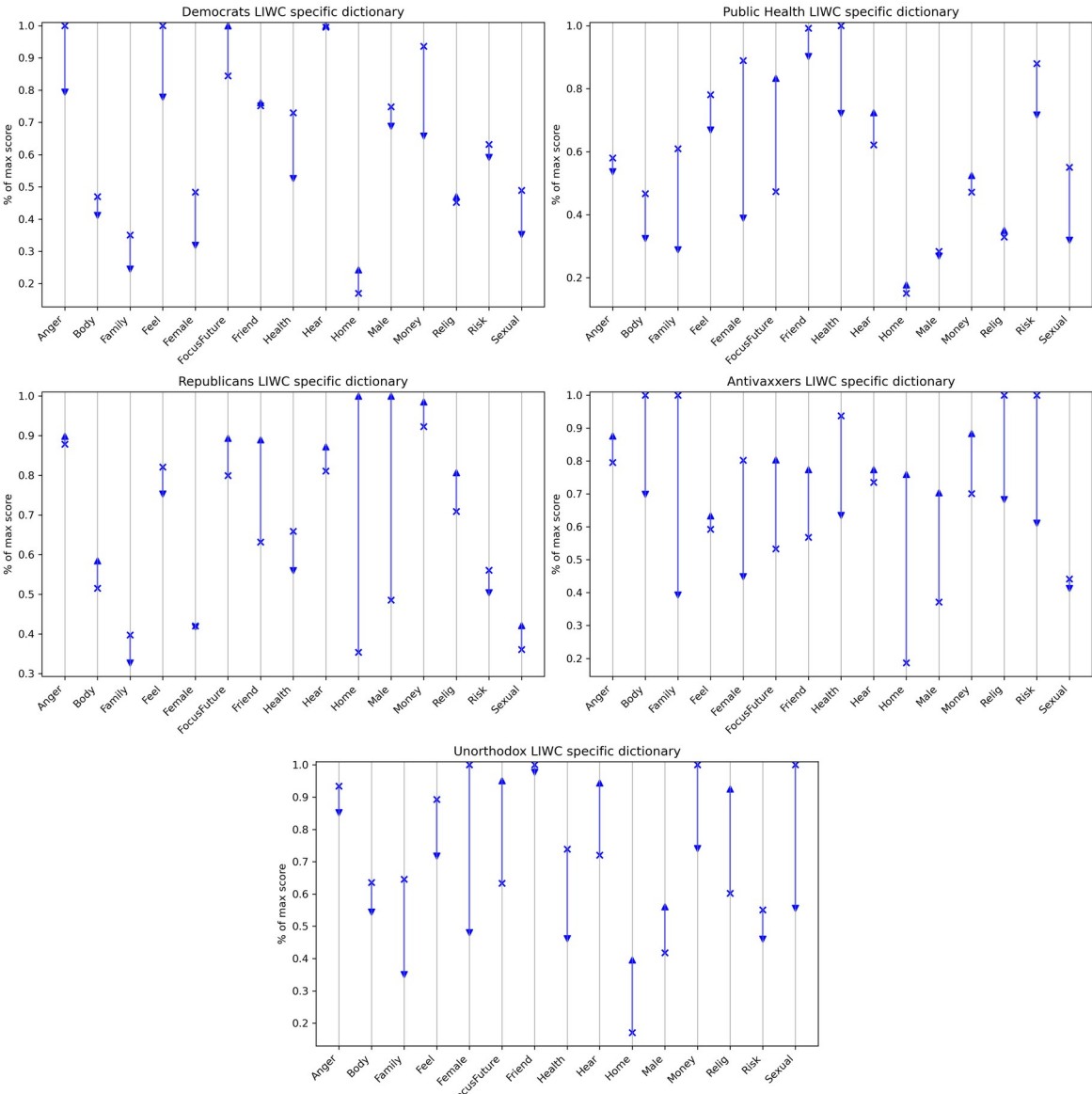

**Fig 5. Changes on selected LIWC components pre- and post-declaration.** X axis shows top 15 absolute score changes. Y axis shows percentage of max pre/post score. Arrow shows direction of change.

became the first state to issue stay-at-home orders on the 19th of March and that 15 further states followed suite over the next five days [47]. In response to these restrictions, a cascade of anti-lockdown protests swept across the United States [48].

**MFT dictionaries.** Using custom Moral Foundations Theory dictionaries, we can further examine the specifically normative discourse each group shows around vaccines. We here discuss selected columns in Fig 6. The full table is available here, or upon request. As with the LIWC dictionaries, the dendrograms in Fig 7 show a linguistic realignment over the course of the study.

Moving along the five moral foundations from pre- to post-declaration, we observe two patterns that reflect the consolidation of an Antivax-Republican alliance at one end, and a Democrat-Public Health partnership at the other. All groups demonstrate a decrease in the

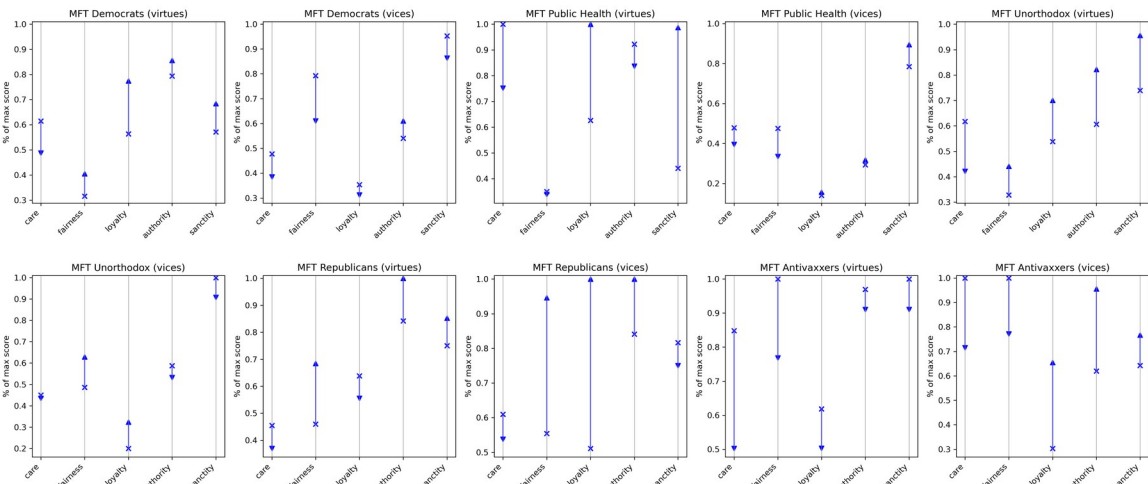

**Fig 6. Changes in score for moral foundations dictionaries pre- and post-declaration.** Components on X axis absolute score changes. Y axis shows percentage of max pre/post score across all groups (for raw scores, see SM §2.2). Arrow shows direction of change.

proportion of care-related words, with the largest decrease among Antivaxxers, who end up converging with Republicans in placing the least emphasis on care. On the loyalty foundation, we see that Democrats and Public Health converge in placing additional weight on the virtue and little emphasis on the vice, while Republicans and Antivaxxers move in the opposite direction and place less emphasis on the virtue and more on the vice. Republicans and Antivaxxers also see a relatively large uptick in emphasis on the vices of authority. Taken together, these results suggest that whereas Democrats and Public Health became increasingly concerned with questions of collective responsibility, Antivaxxers and Republicans were less concerned with collective well-being and became progressively more antagonistic towards state and federal authorities. This interpretation dovetails a recent study by [49], which finds that conservatism and anti-vaccination attitudes are both strongly correlated with opposition to public health interventions.

Whereas most communities changed relatively little along the fairness foundation, it is worth noting that Republicans and Antivaxxers again show considerable convergence on both

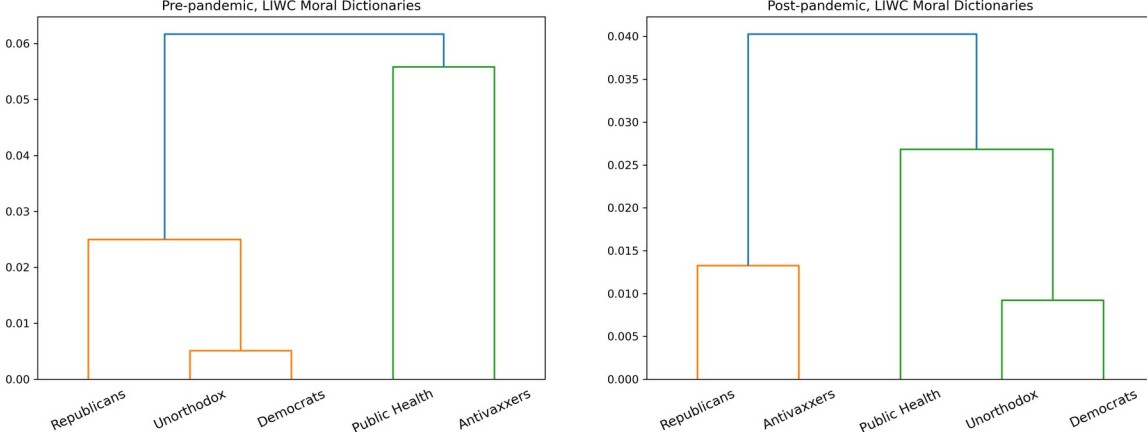

**Fig 7. Dendrogram for custom MFT virtue dictionaries.**

the vice and virtue dimension of this foundation post-declaration. Finally, with respect to sanctity, the virtue and vice dictionaries tend to move in opposite directions. Only the Public Health community increases for both virtue and vice.

In sum, these trends evidence a willingness among but not across both sets of communities to adjust their initial moral emphasis so as to accommodate the values of their closest interlocutors; indicating that the discourse evolved in line with pre-existing patterns of social connection. Interestingly, these dynamics of moral (dis)agreement depart somewhat from traditional models of (de)polarization. While it is true that communities situated at opposing ends of the spectrum move further apart, those who initially emphasize just a few shared foundations seem ready to negotiate their remaining moral differences. Thus, while polarization between poles persists, within each pole, depolarization takes place. On the one hand, this pattern suggests that moral compromise is possible; on the other, it also suggests that local compromise with your closest allies can lead to increased global disagreement [50].

## RQ3: Social mechanisms for polarization

The reasons for convergence are likely to be complex and various: all groups are responding both to one another and to ongoing offline events as the pandemic unfolds. The use of VAR sheds light on one set of patterns.

The VAR analysis showed several significant coefficients at $p \leq 0.05$ (uncorrected), pictured on the left side of Fig 8. Each coefficient represents the predicted increase in influence of a group at time $t$ given a 1-retweet increase by some group at time $t - 1$. Tests for Granger causality indicate that all and only the depicted arrows are Granger-causal relationships. Since the number of tweets and influential retweets varies substantially across the groups, the right side of Fig 8 shows the coefficients multiplied by the total influence of the inbound group across the whole time series, presented on a log scale for comparison.

The VAR model without exogenous variables shows several interesting patterns of interaction. There are reciprocal interactions between both Republicans and Antivaxxers, and between Democrats and Public Health accounts. While some of the coefficients seem small, the normalized net influence shows that (e.g.) the influence of Antivaxxers on Republicans was comparable to that of Republican self-promotion. So a first important lesson from the time series seems to be that the convergence between groups was driven by the interaction between them, rather than by independent evolution.

Including public health measures as an exogenous variable adds further nuance to the picture. Fig 9 shows the results of incorporating public health measures as an exogenous variable.

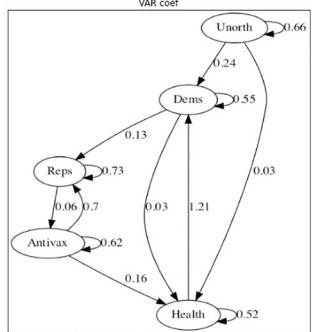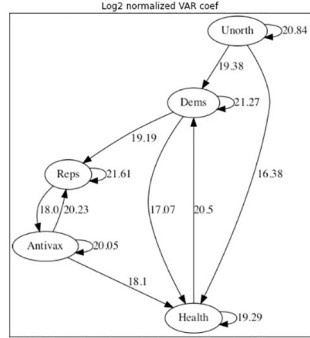

**Fig 8. Significant VAR coefficients for the simple lag-1 model.** The left side shows the raw coefficients, the right side shows the log-normalized influence taking into account total influence of each group (see text for details).

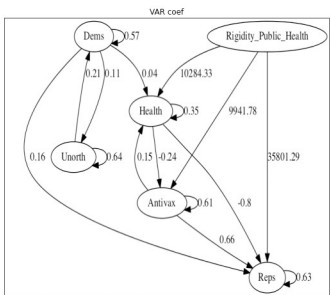
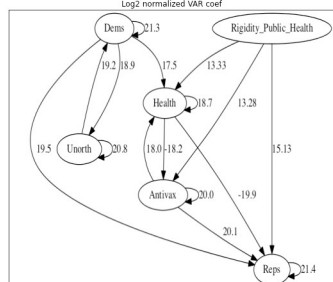

**Fig 9.** Left: significant VAR coefficients for a model incorporating public health measure. Right: log-normalized influence.

This is an aggregate measure, focusing on the United States, that includes things such as mask-mandates and stay-at-home orders that were implemented in the weeks following the pandemic declaration.

Treating public health measures as an exogenous variable preserves some structural relationships—notably, the influence of Democratic users seems to be largely independent of public health measures. On the other hand, Republicans and Antivaxxers (and, less surprisingly, Public Health) gain in influence as public health measures get stronger. The influence of Public Health officials on Republicans and Antivaxxers also switches to strongly negative, suggesting that their messaging was effective in context. Finally, we note that the relationship between Antivaxxers and Republicans becomes asymmetric: once we take public health measures into account, Antivaxxers increase the influence of Republican messaging but not vice-versa.

## Discussion

### Summary of results

To answer **RQ1** we used modularity clustering, an unsupervised method that consistently partitioned the communities into Democrats, Republicans, Public Health, Antivaxxers, and Unorthodox. We also justified the methodology and provided a detailed qualitative and quantitative description of each group. In order to answer **RQ2** we used Linguistic Inquiry and Word Count (LIWC) to study the moral and non-moral language used by these communities. Our analysis of both the linguistic and network behavior of these communities shows that two distinct polarized axes—Democrat-Republican and Health-Antivax—converged into a single polarized discourse around vaccination. We call this 'convergent polarization.' This polarization is now firmly entrenched as part of the US political landscape [36, 38, 51, 52]. Our response to **RQ3** partly explains why this might be.

Recall that to address **RQ3** we used vector autoregression analysis (VAR) with the retweet rate of each of the communities as endogenous variables, and public health interventions as the exogenous variable. Our results show that responsiveness to public health measures varied across groups and can explain part of the polarization we observed. Yet the question remains as to why the groups responded differently, which requires reflecting on dynamics of political engagement. On the basis of our observations, we here examine the *value-first* and the *trust-first* dynamics, and offer evidence in support of the latter.

It is important to keep in mind that things could have been different. One could imagine (for example) both sides of the political spectrum coming together in opposition to antivaxxers; this was largely the experience in Australia, as well as several European countries [53, 54]. Polarization could have also gone the other way; anti-vaccination groups have, after all, often had a left-

leaning component [55]. Given that Donald Trump was spearheading a vaccination push, it would not have been surprising for his followers to rally around him [34, 38]. Finally, it could have been the case for most groups to have intermediate and cross-cutting concerns, as appears to have happened with the Unorthodox community. Yet none of this happened. Why?

One possibility is that shifts in signal-boosting are driven by a *values-first* dynamic, in which a group is treated as an ally worthy of signal-boosting if and only if they tend to publish information that expresses values shared with your group. In previous work, Haidt and colleagues have found that more politically conservative people and communities tend to place greater emphasis than liberals or leftists do on the so-called 'binding' foundations of loyalty, authority, and sanctity [56]. In addition, political conservatives are typically found to be higher in dispositional disgust-sensitivity, which is associated with the sanctity foundation [57] (though see [58] for a dissenting view).

When it comes to COVID-19 and vaccination in particular, it has been hypothesized that normative health behaviors during the pandemic may be partially explained by individual differences in moral foundations. Americans who score high on the care and fairness domains are more likely to report staying at home, support wearing face-masks, and respect social distancing, while those who score high on the sanctity domain are more likely to report wearing face-masks and comply with social distancing, but less likely to limit their movement [59]. Parents who score high on the sanctity foundation are especially likely to be fence-sitters or rejecters when it comes to vaccinating their children [60]. Parents high on the sanctity foundation, low on the authority foundation, and high on the care foundation are more likely to be rejecters [3].

Our data offers only equivocal support for these hypotheses. At least in the context of vaccine discourse, Democrats and Public health appear to score high on (e.g.) sanctity, while Republicans became much more concerned with fairness over the course of the pandemic. Indeed, one of the striking findings of our work is the degree to which groups were willing to change both their moral and non-moral language-use around vaccines over the course of the pandemic. Values, at least to the extent that they can be extracted by linguistic analysis, changed with the emergence of COVID-19 and did not constitute the fix point that explains the polarization.

An alternative hypothesis is that signal-boosting is driven by a *trust-first* dynamic: groups are treated as allies worthy of signal-boosting if and only if your group has reasons to trust them, regardless of informational veracity. On a trust-first dynamic, information from trusted sources can push one to update what you believe and your commitment to particular values, precisely because you trust your sources [61]. Trust-first dynamics are susceptible to what Begby [62] calls 'evidential pre-emption', which occurs when a trusted source also warns that one is likely to encounter misleading contrary evidence. Trust has been theorized as an unquestioning attitude towards testimony [63]. While not intrinsically bad, evidential pre-emption arguably plays an important role in spreading conspiracy theories [15, 64] and supporting online echo chambers [65].

Our data are consistent with a trust-first dynamic. One possible force pulling together Republicans and Antivaxxers is a shared distrust of scientific expertise [51, 52]. Republican mistrust of the scientific establishment dates back at least as far as the 1980s [66], steadily increased in the context of climate change [67], and has become even more pronounced in response to COVID-19 [51, 68]. Furthermore, while conservative self-identification is correlated with vaccine hesitancy, this appears to be mediated by distrust of scientific expertise, vanishing when distrust is controlled for [5]. In contrast to these pre-existing patterns of distrust, Democrats have historically scored high on trust in science [69]. Moreover, and as we would expect if our trust-first hypothesis is correct, Democrats have become more confident in medical experts over the course of the pandemic [70].

Patterns of trust might also shed some light on the intermediate position of the Unorthodox group. Recall that a primary concern of this group was unease at vaccine trials in Africa. There is a well-established pattern of distrust of medical experimentation on Blacks, even on the left, stemming from historical abuses such as the Tuskegee syphilis experiment [71]. It is important to bear in mind that there is no evidence that this community opposes vaccines *tout court*, or subscribes to a conservative world view. Instead, their distrust for vaccination seems rooted in concerns with issues of colonial and racial injustice. So while partisan attitudes about trust in science might be overall entrenched, the Unorthodox group shows that specific patterns of distrust can cross-cut these broader patterns.

Our explanation coheres with the hypothesis advanced by DiResta & Lotan [72, 73], who argue that Twitter's content moderation aimed at medical misinformation unified Antivaxxers and Republicans because it prompted Antivaxxers to *reframe* their message. Rather than straightforward medical misinformation (e.g., 'Vaccines cause autism'), Antivaxxers began to seek political cover by allying with Republicans and reframing their message in political terms (e.g., 'Mandatory vaccination is tyranny'). Within the context of COVID-19 therefore, both Twitter and Public Health authorities represented a shared focus of distrust, against which resistance would seem appropriate for some groups.

## Limitations and future work

As with all observational work on Twitter, our data collection was limited by what Twitter makes available. Roughly 1% of all tweets [18] are available to researchers without a commercial contract. Our corpus is also mostly in the English language, which we estimate represents 68% of Twitter chatter worldwide pertaining to COVID-19. Future work to address these issues may include augmenting our corpus with multi-lingual and more recent COVID-19 Twitter data sets. Recent work on estimating the distribution of missing tweets from the API stream [74] may also offer a more nuanced picture of any gaps.

As a purely observational study, we are limited in the sorts of causal conclusions we can draw. That said, our work does suggest avenues for future research. We used a proxy measure for public health interventions that aggregated at the level of the entire US. Since many public health measures were implemented state-by-state, coordinating twitter's (sparse) geolocation data with data about local interventions might shed further light on the influence of local public health measures. Indeed, a recent study by [75] shows that there are important regional differences in partisan responsiveness to public health mandates.

Our data set was limited to tweets which discussed vaccines. An important open question therefore is whether the dynamics of trust that we identify extend beyond vaccine-related discourse. Looking for similar dynamics in different domains, or coordinating distinct actors across different topics, might shed further light on these questions.

Finally, we relied on dictionaries specific to Moral Foundations Theory to make the case for a shift in values. Despite enjoying widespread adoption within academia and even crossing the mainstream into popular culture (e.g. [76, 77]), Moral Foundations Theory has also attracted sustained criticism [78, 79]. Future work might consider other taxonomies of moral reasons, such as Morality as Cooperation [80], or consider using data-driven approaches (such as sentiment analysis) to tease apart the valence of different responses.

## Conclusion

In this paper, we showed how the online conversation about vaccines underwent a political realignment as the COVID-19 pandemic progressed. By analyzing networks, language use, and time series data, we demonstrated a realignment of parties around a familiar, politically

polarized, set of axes. While these dynamics are largely entrenched by now, they were in flux in the early months of the pandemic. Hence, our research provides a unique window into their formation.

We have argued that this realignment shows that the dynamics of online discourse are driven by social connection and trust, rather than by underlying static values. If so, then trusted informants might be uniquely well positioned to change peoples beliefs. Hence, rather than target hesitant communities with evidence that they are likely to ignore, political resources could be used to identify individuals who are trusted by the target group. Recent reports suggest that this strategy is already gaining traction, at least in the United States, where the Biden administration has partnered with social media 'influencers' in an effort to promote vaccination among the countries younger population [81]. Previous work has suggested that personal accounts play an important role in both pro- and anti-vaccination groups on twitter [10]. Furthermore, although *reframing* was part of the problem, it could be used to signal pro-vaccination messages in terms that are attractive to dissenters: 'vaccines are good for business', or 'protect your body with vaccines'. More recent work on moral and political persuasion has also suggested that personal narratives are more convincing than purely factual accounts precisely because they help build interpersonal trust [82]. Vaccination is a critical public health issue, and disseminating accurate information is vital. Whether accurate information is believed and acted upon, however, is a matter of interpersonal trust—and it is here, we suggest, that there remains substantial work to be done.

## Supporting information

**S1 Appendix. Supplementary information on the data and methods.** This includes data collection considerations (incl. hashtags), analysis of bots in the dataset, biases in data, classification tasks, and structural break analyses.
(ZIP)

## Author Contributions

**Conceptualization:** Ignacio Ojea Quintana, Ritsaart Reimann, Mark Alfano.

**Data curation:** Ignacio Ojea Quintana, Marc Cheong, Colin Klein.

**Investigation:** Ritsaart Reimann, Mark Alfano, Colin Klein.

**Methodology:** Ritsaart Reimann, Marc Cheong.

**Project administration:** Ignacio Ojea Quintana.

**Resources:** Marc Cheong, Colin Klein.

**Software:** Ignacio Ojea Quintana, Mark Alfano, Colin Klein.

**Supervision:** Ignacio Ojea Quintana, Colin Klein.

**Visualization:** Ignacio Ojea Quintana.

**Writing – original draft:** Ignacio Ojea Quintana, Ritsaart Reimann.

**Writing – review & editing:** Ignacio Ojea Quintana, Ritsaart Reimann, Colin Klein.

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
