## [Decision Letter · Decision Letter 0]

23 May 2022

PONE-D-21-40016Polarization and trust in the evolution of vaccine discourse on Twitter during COVID-19PLOS ONE

Dear Dr. Ojea Quintana,

Thank you for submitting your manuscript to PLOS ONE. After careful consideration, we feel that it has merit but does not fully meet PLOS ONE’s publication criteria as it currently stands. Therefore, we invite you to submit a revised version of the manuscript that addresses the points raised during the review process.

We look forward to receiving your revised manuscript.

Kind regards,

Tingshao Zhu

Academic Editor

PLOS ONE

Journal Requirements:

2. Please ensure that you refer to Figure 6 in your text as, if accepted, production will need this reference to link the reader to the figure.

Additional Editor Comments (if provided):

Reviewers' comments:

Reviewer's Responses to Questions

**Comments to the Author**

1. Is the manuscript technically sound, and do the data support the conclusions?

Reviewer #1: Yes

Reviewer #2: Yes

Reviewer #3: Yes

Reviewer #4: Yes

2. Has the statistical analysis been performed appropriately and rigorously? 

Reviewer #1: Yes

Reviewer #2: Yes

Reviewer #3: Yes

Reviewer #4: Yes

3. Have the authors made all data underlying the findings in their manuscript fully available?

Reviewer #1: Yes

Reviewer #2: Yes

Reviewer #3: Yes

Reviewer #4: Yes

4. Is the manuscript presented in an intelligible fashion and written in standard English?

Reviewer #1: Yes

Reviewer #2: Yes

Reviewer #3: Yes

Reviewer #4: Yes

5. Review Comments to the Author

Reviewer #1: The article has been written meticulously and addresses an important public health issue of developed and developing nations alike using an innovative approach. In my opinion all the findings and interpretations are scientifically sound as presented in the paper.

Reviewer #2: The manuscript is written in very prominent contemporary debate. Scientifically written in logical flows. "We have argued that this realignment shows that the dynamics of online discourse are driven by social connection and trust, rather than by underlying static values." Agreed.

Reviewer #3: The authors describe an analysis of the debate about COVID-19 on Twitter over the first five months of the pandemic. The final aim is to study English-language discourse around vaccines, vaccination-related engagement and discourse, and the causes of the changes in engagement. They describe the data collected, their approach, and the observations derived from the analyses. The authors focus their research on the top five communities they extracted (as they represent most of the English-language data they collected), summarising their statistics, and focusing all of their analyses on these groups. The authors provide detailed descriptions of the outcomes of their study by answering their research questions and properly discussing the limitations of their approach.

The article is well organised, and the outcomes are discussed thoroughly. I advise providing additional context and explanations for the following points.

- In the data collection part, when mentioning “...a series of vaccination-related keywords, hashtags, and short expressions…”, it would be good to have a few examples of such words/expressions (even if they are included in the Appendix) with a small discussion on how and why such words were chosen (maybe you could pick the most relevant and/or complex ones).

- The authors discarded “Users that only retweeted but never authored an original tweet” while keeping “...a very small number of bots”. Both users with retweets only and bots may play a signal-boosting function. Discussing the reasons behind the decision of keeping one category while discarding the other may improve the soundness of the article. Moreover, how different would the network be without bots’ data? Do bots really change the shape of the network or the interactions between the communities?

- The authors considered only the top five communities, mainly because they share content using the English language while representing 80% of the nodes and 90% of the tweets. How and why did the authors choose only the top five? What is the choice criterion? Was it only because of the language used within the identified communities?

- The authors state “Communities beyond the top five also tended to cluster around non-English-language accounts, which limit the utility of our dictionary-based tools”. Did the authors discard all the non-English content the top five communities shared? I advise discussing such a point and carrying out a quick analysis of the percentages of English content VS non-English for the top five (or even 10) groups to cover such an aspect, also providing further support for the discussion about the choice of the top five communities.

Furthermore, the authors mention the Appendix many different times within the text, although no Appendix is provided within the article.

Reviewer #4: 1. The article is very clear and well defined. But still some corrections are presents.

2. Authors need to focus the novelty in abstract section and expected output with actual output.

3. introduction section is too short to intro the proposed work motto.

4. Result section is very organized but images are unclear. Need to upgrade with high resolution images.

5. Try to use some recent papers as related work (2020-2022).

6. There are some grammatical correction, please recheck and correct.

6. PLOS authors have the option to publish the peer review history of their article (what does this mean?). If published, this will include your full peer review and any attached files.

Reviewer #1: **Yes: **Dr. Aftab Ahmad, MD (Community Medicine)

Reviewer #2: No

Reviewer #3: **Yes: **Andrea Tocchetti

Reviewer #4: No

---

## [Author Response · Author response to Decision Letter 0]

4 Jun 2022

Please see the attached letter in response to reviewers, where we address each of the helpful comments in detail.

---

## [Decision Letter · Decision Letter 1]

29 Jul 2022

PONE-D-21-40016R1Polarization and trust in the evolution of vaccine discourse on Twitter during COVID-19PLOS ONE

Dear Dr. Ojea Quintana,

Thank you for submitting your manuscript to PLOS ONE. After careful consideration, we feel that it has merit but does not fully meet PLOS ONE’s publication criteria as it currently stands. Therefore, we invite you to submit a revised version of the manuscript that addresses the points raised during the review process.

Please see reviewer comments below.

We look forward to receiving your revised manuscript.

Kind regards,

Hanna Landenmark

Staff Editor, PLOS ONE

on behalf of 

Tingshao Zhu

Journal Requirements:

Additional Editor Comments (if provided):

Reviewers' comments:

Reviewer's Responses to Questions

**Comments to the Author**

1. If the authors have adequately addressed your comments raised in a previous round of review and you feel that this manuscript is now acceptable for publication, you may indicate that here to bypass the “Comments to the Author” section, enter your conflict of interest statement in the “Confidential to Editor” section, and submit your "Accept" recommendation.

Reviewer #3: All comments have been addressed

Reviewer #4: All comments have been addressed

Reviewer #5: All comments have been addressed

Reviewer #6: (No Response)

Reviewer #7: (No Response)

2. Is the manuscript technically sound, and do the data support the conclusions?

Reviewer #3: Yes

Reviewer #4: Yes

Reviewer #5: No

Reviewer #6: Yes

Reviewer #7: Yes

3. Has the statistical analysis been performed appropriately and rigorously? 

Reviewer #3: Yes

Reviewer #4: Yes

Reviewer #5: No

Reviewer #6: Yes

Reviewer #7: Yes

4. Have the authors made all data underlying the findings in their manuscript fully available?

Reviewer #3: Yes

Reviewer #4: Yes

Reviewer #5: No

Reviewer #6: Yes

Reviewer #7: Yes

5. Is the manuscript presented in an intelligible fashion and written in standard English?

Reviewer #3: Yes

Reviewer #4: Yes

Reviewer #5: No

Reviewer #6: Yes

Reviewer #7: Yes

6. Review Comments to the Author

Reviewer #3: The authors presented a study analysing the impact of COVID-19 on the vaccine discourse over the first 5 months of pandemic on Twitter. They clearly stated the research questions they address and provide detailed analyses covering such questions. The authors properly describe the methods, data, and procedures employed in the process.

Reviewer #4: Authors are successfully addressed all of the comments from reviewer. No further corrections are identified.

Reviewer #5: The topic presented by the authors is very interesting and timely. However, there are several major issues that need to be addressed.

1. The abstract is very vage and not very clear. Please revise in a way the reader knows exactly what to expect from the paper.

2. I like highlighting RQ1 - RQ3 on page 2. However, I suggest to be more specific.

RQ1. Which groups are most important in the English-language discourse around 40 vaccines on Twitter? I do not think that is what the authors do. Instead, the focus is on specific countries using the English language. Which countries are these? Add to RQ1!

RQ2. How did vaccination-related engagement and discourse change over the first 42 five months of the pandemic? Add years, ie. 02/2000 - 07/2000 or similar.

RQ3. What social forces might help explain observed changes in engagement? should be

RQ3. What social forces might help in explaining observed changes in engagement?

In general, the language needs to be improved of the paper. In its current form it is very poor language at many places and I cannot list all of them.

As explained in the sections coming (just another example of poor language).

3.To identify different communities, we used Gephi’s implementation of the Louvain 110 modularity algorithm developed by [22] This is unclear. Please discuss community/module detection algorithms in details, e.g.,

https://link.springer.com/article/10.1186/s12859-016-0979-8

4. Discussion: From the beginning it is unclear to me how to define 'Antivaxxers'? I searched the paper but could not find a proper definition (it should pop out immediately).

Question: Is the population of a country with a low vaccination rate (e.g. some African countries) in the category Antivaxxers? I is unclear to me if this is a negative term or neutral and if one has a choice to be in this group or is this possible determined by a third party.

In this context, it would be good to add some English-speaking African countries with a low vaccination rate and low death rate which I think exist. If the focus is only on the US a discussion of this would be sufficient.

Considering this the distinction Health-Antivax seem incorrect. I would suggest a revision of wording to make the presentation more factual.

4. From line 407 to end: I find the connection between the numerical results and the provided discussion unsatisfactory. The main problem is a lack of connection. The authors do not use the numerical results to provide a discussion but the discussion seems not well grounded at all. At least I could not see such a connection.

It is very important that the discussion focuses exclusively on the interpretation of the numerical results. At the end of the discussion, a wider perspective, which could be even speculative, could be provided.

I suggest to rewrite the entire discussion section.

5. Our corpus is also mostly in the English language, 481 which we estimate represents 68% of Twitter chatter worldwide pertaining to COVID. How can this information be relevant when Democrat-Republican are limited to the US only?

Similar to the point about Health-Antivax (see 4) also

Democrat-Republican is not well defined. Definitions of all 4 terms should be added to the methods section.

6. The conclusion section is similar unclear and lacks a connection to the numerical results.

Reviewer #6: This paper presents an original and as of yet unpublished study. The authors have clearly and effectively described their analyses and drawn appropriate (and quite interesting) conclusions from the analyses presented. The article is also well written and understandable, with one exception which stood out to me while reading the manuscript.

The dataset obtained from the Twitter Streaming API appears to be global, with no mention of any geographic restrictions from users, and several non-US users are listed, with the particular grouping of the "unorthodox" being non-American in nature. However, two of the communities are defined in terms of American political divisions, the public health measure is US-centered, and geolocation data are mentioned in the discussion section when addressing future work. I think it may be helpful to clarify briefly but explicitly either in the data collection subsection or in the community characterization subsection that, even though the labels "Republican" and "Democrats" are used for two communities, the data extend globally. As a secondary, optional, matter I would be interested to know why American political cleavages may define a global discussion space.

Reviewer #7: This paper reports an interesting analysis of the discourse on vaccines around the onset of the COVID-19 pandemic. In particular, it shows how the discourse changed after the WHO officially declared the pandemic in March 2021. Overall, I think that it is well done and makes a significant contribution.

Although I review the paper for the first time, I see that it is a resubmission. Therefore, I limit my critical comments to points that can be reasonably addressed at this stage of the review process.

First, I encourage the authors to clarify the geographic scope of the analysis. The dataset includes only tweets written in English, but is not restricted to particular countries. At the same time, the analysis relies on the categories of US politics, particularly the distinction between Republicans and Democrats. This tension should be clarified as early as possible in the paper.

Second, the link between vaccines and COVID was a bit confusing to me, given the time frame of the analysis (December 2019-June 2020). In that period, the emphasis was not on COVID vaccines, but more on the nature of the disease and mitigation strategies such as masks and containment. The authors do frame the study as an analysis of changes of the vaccination discourse following the emergence of COVID, but they could be more explicit that the discourse is on vaccines _in general_. Currently, readers may have the wrong expectation that the paper is about COVID vaccines.

Third, the pre-post comparisons are the most interesting parts of the analysis, but they could be discussed more in depth. For example, we see in Figure 1 that the structure of the retweet network changes significantly, but the point is barely elaborated in the text.

Finally, Figures 5 and 7 are not legible, partly because of the low resolution but also due to the lack of a legend and the small size of the points. These figures should be improved.

7. PLOS authors have the option to publish the peer review history of their article (what does this mean?). If published, this will include your full peer review and any attached files.

Reviewer #3: **Yes: **Andrea Tocchetti

Reviewer #4: **Yes: **F.M. Javed Mehedi Shamrat

Reviewer #5: No

Reviewer #6: No

Reviewer #7: No

---

## [Author Response · Author response to Decision Letter 1]

15 Aug 2022

Please find attached a version of the document copied below.

Dr Ignacio Ojea Quintana and coauthors

Australian National University

Canberra

ACT 2600

Australia

August 10th, 2022.

Re: Response to Reviewers, PLOS ONE / PONE-D-21-40016

Dear Hanna Landenmark and Tingshao Zhu, on behalf of the Editors and Reviewers,

We thank you, the Editorial Board, and the panel of Reviewers for your generous time in the review process of our paper and for providing us with helpful comments on our manuscript, “Polarization and trust in the evolution of vaccine discourse on Twitter during COVID-19”.

On the following pages, please find our detailed responses to address the comments raised by the reviewer panel.

Kind regards,

Dr Ignacio Ojea Quintana

on behalf of all coauthors 

(Ignacio Ojea Quintana, Ritsaart Reimann, Marc Cheong, Mark Alfano, Colin Klein)

Encl. (Responses to the Review Report)

Responses to the Review Report

Reviewer #3:

The authors presented a study analysing the impact of COVID-19 on the vaccine discourse over the first 5 months of pandemic on Twitter. They clearly stated the research questions they address and provide detailed analyses covering such questions. The authors properly describe the methods, data, and procedures employed in the process.

- We thank the reviewer for their charitable comments.

Reviewer #4: 

Authors are successfully addressed all of the comments from reviewer. No further corrections are identified. (sic.)

- We are glad to have addressed the comments, and we thank the reviewer for them since they genuinely improved the essay.

Reviewer #5: 

The topic presented by the authors is very interesting and timely. However, there are several major issues that need to be addressed.

We thank the reviewer for their comments.

In the next paragraphs we will address them one by one.

1. The abstract is very vague and not very clear. Please revise in a way the reader knows exactly what to expect from the paper.

- The abstract was modified to improve clarity. It now reads:

“Trust in vaccination is eroding, and attitudes about vaccination have become more polarized. This is an observational study of Twitter analyzing the impact that COVID-19 had on vaccine discourse. We identify the actors, the language they use,how their language changed, and what can explain this change.

First, we find that authors cluster into several large, interpretable groups, and that the discourse was greatly affected by American partisan politics. Over the course of our study, both Republicans and Democrats entered the vaccine conversation in large numbers, forming coalitions with Antivaxxers and public health organizations, respectively. After the pandemic was officially declared, the interactions between these groups increased. Second, we show how the moral and non-moral language used by the various communities converged in interesting and informative ways. Finally, vector autoregression analysis indicates that differential responses to public health measures are likely part of what drove this convergence. Taken together, our results suggest that polarization around vaccination discourse in the context of COVID-19 was ultimately driven by a trust-first dynamic of political engagement.”

The abstract states our three main contributions: (i) an observational study of which communities were identified, (ii) an analysis of how their language changed, and (iii) an explanatory hypothesis using vector autoregression, complemented with another hypothesis about what drove the dynamics.

In the body of the paper we also made an effort to be more precise in the description of the techniques used and the numerical results. But since all other reviewers found the original abstract sufficiently clear, we do not want to make substantial changes to the original version.

2. I like highlighting RQ1 - RQ3 on page 2. However, I suggest to be more specific.

RQ1. Which groups are most important in the English-language discourse around vaccines on Twitter? I do not think that is what the authors do. Instead, the focus is on specific countries using the English language. Which countries are these? Add to RQ1!

RQ2. How did vaccination-related engagement and discourse change over the first five months of the pandemic? Add years, ie. 02/2000 - 07/2000 or similar.

RQ3. What social forces might help explain observed changes in engagement? should be

RQ3. What social forces might help in explaining observed changes in engagement?

- We thank the reviewer for asking us to sharpen our research questions. With respect to RQ2 and RQ3, the suggested changes have been incorporated. With respect to RQ1, we have added some comments to clarify that we did not limit data collection to the United States nor any other territory, and that the prevalence of specific regions within our dataset is an artifact of the interaction between our methodology and the distribution of Twitter’s user-base. In particular, we point out that the prevalence of U.S-based users within our analysis is partly due to our focus on English-language discourse; partly due to the fact that the vast majority of Twitters English speaking user-base is located in the United States; and partly a reflection of the extent to which American discourse defines global online discussions, at least when those discussions are carried out in English.

In general, the language needs to be improved of the paper. In its current form it is very poor language at many places and I cannot list all of them.

- As explained in the sections coming (just another example of poor language).

In the next few responses we explain thow we clarified the language used and the structure of the essay in a way that is clearly tied with the observations we made.

3.To identify different communities, we used Gephi’s implementation of the Louvain 110 modularity algorithm developed by [22] This is unclear. Please discuss community/module detection algorithms in details.

- Many thanks for this comment.

The Materials and Methods section now includes a better explanation of what modularity is and the algorithm used.

In fact, in the original version of the essay we provided some description but we decided to remove it for the first R&R, so we are sympathetic to the comment of the reviewer. We now included a description in some detail of how the unsupervised method works. Nevertheless, we decided not to present the mathematical and algorithmic aspects in excessive detail because we regard the method as standard in the literature, and we do not want the readers to get stuck in unnecessary details.

4. Discussion: From the beginning it is unclear to me how to define 'Antivaxxers'? I searched the paper but could not find a proper definition (it should pop out immediately).

Question: Is the population of a country with a low vaccination rate (e.g. some African countries) in the category Antivaxxers? I is unclear to me if this is a negative term or neutral and if one has a choice to be in this group or is this possible determined by a third party.

In this context, it would be good to add some English-speaking African countries with a low vaccination rate and low death rate which I think exist. If the focus is only on the US a discussion of this would be sufficient.

Considering this the distinction Health-Antivax seem incorrect. I would suggest a revision of wording to make the presentation more factual.

Similar to the point about Health-Antivax also, Democrat-Republican is not well defined. Definitions of all 4 terms should be added to the methods section.

- We thank the reviewer for asking us to more carefully define each group, and antivaxxers in particular. To avoid confusion, we have added a definition of ‘Antivaxxers’ to the introduction, specifying that we use this term in a descriptive rather than normative sense: it is our interpretation of a community of users whose top hashtags and accounts display anti-vaccination attitudes. We also note that whether or not users belong to this cluster is not contingent on their geographic location; for as explained in the Network construction and community clustering section, community membership is determined by social interaction (e.g., retweeting). 

The distinctions between Democrats and Republicans on the one hand and Antivaxxers and Public Health organizations on the other are addressed in more detail in the ‘community characterization and classification’ section. We emphasize that since we are dealing with large and diverse populations of users about whom we only have limited information, our definitions are informed by broad patterns of similarities and differences within and between the various communities.

With respect to the U.S focused nature of our analysis, we have added a section to clarify that we did not limit data collection to the United States, and that the prevalence of American politics within our analysis is an artifact of the interaction between our methodology and the distribution of Twitter’s English-speaking user-base. 

Finally, with respect to the reviewers request to add English-speaking African countries, we note that these are already included in the original paper. In particular, our analysis of the unorthodox community makes it clear that there are African-based users involved in the discourse, and that these users express both pro- and anti-vaccination attitudes.

5. From line 407 to end: I find the connection between the numerical results and the provided discussion unsatisfactory. The main problem is a lack of connection. The authors do not use the numerical results to provide a discussion but the discussion seems not well grounded at all. At least I could not see such a connection.

It is very important that the discussion focuses exclusively on the interpretation of the numerical results. At the end of the discussion, a wider perspective, which could be even speculative, could be provided. I suggest to rewrite the entire discussion section.

- The discussion section is now reorganized and rewritten in light of the reviewer’s comments. Two notes on this.

First, we sympathize with the reviewer's objection that some of what we said there was not transparently tied with the observational results that we presented before. We now organize the material in a way that is justified by the results. In particular, our discussion of the values-first vs trust-first dynamics is warranted by the observations we did in our linguistic analysis, in particular the use of moral language by different communities.

Second, the purpose of the discussion section is to provide a broader theoretical background in order to explain the observations made. For this reason, that section builds on and discusses some contemporary literature.

6. Our corpus is also mostly in the English language, which we estimate represents 68% of Twitter chatter worldwide pertaining to COVID. How can this information be relevant when Democrat-Republican are limited to the US only?

- We now clarify this important point in the essay.

The prevalence of U.S-based users within our analysis is partly due to our focus on English-language discourse; partly due to the fact that the vast majority of Twitter’s English speaking user-base is located in the United States (68%); and partly a reflection of the extent to which American discourse defines global online discussions, at least when those discussions are carried out in English. This explains the brute fact that random sampling using English words will most likely give an over-representation of USA discourse and dynamics, which is what we found.

Reviewer #6: 

This paper presents an original and as of yet unpublished study. The authors have clearly and effectively described their analyses and drawn appropriate (and quite interesting) conclusions from the analyses presented. The article is also well written and understandable, with one exception which stood out to me while reading the manuscript.

The dataset obtained from the Twitter Streaming API appears to be global, with no mention of any geographic restrictions from users, and several non-US users are listed, with the particular grouping of the "unorthodox" being non-American in nature. However, two of the communities are defined in terms of American political divisions, the public health measure is US-centered, and geolocation data are mentioned in the discussion section when addressing future work. I think it may be helpful to clarify briefly but explicitly either in the data collection subsection or in the community characterization subsection that, even though the labels "Republican" and "Democrats" are used for two communities, the data extend globally. As a secondary, optional, matter I would be interested to know why American political cleavages may define a global discussion space.

- We thank the reviewer for raising this point, and have clarified that despite identifying two clearly U.S. based clusters of users, our data extend globally. This point is made briefly in the introduction, and then elaborated in the community characterization and classification section. We note that since we did not limit data collection to the U.S. nor any other territory, the prevalence of American politics within our analysis is partly due to our focus on English-language discourse; partly due to the fact that the vast majority of Twitters English speaking user-base is located in the United States; and partly a reflection of the extent to which American political cleavages define global online discussions, at least when those discussions are carried out in English. Hence, even though we distinguish 'Democratic' and 'Republican' clusters of users, it is not the case that all users in our data set are based in the United States, for neither Public Health Institutions nor Antivaxxers are unique to U.S. discourse. 

Reviewer #7: 

This paper reports an interesting analysis of the discourse on vaccines around the onset of the COVID-19 pandemic. In particular, it shows how the discourse changed after the WHO officially declared the pandemic in March 2021. Overall, I think that it is well done and makes a significant contribution.

Although I review the paper for the first time, I see that it is a resubmission. Therefore, I limit my critical comments to points that can be reasonably addressed at this stage of the review process.

- We thank the reviewer for their comments.

First, I encourage the authors to clarify the geographic scope of the analysis. The dataset includes only tweets written in English, but is not restricted to particular countries. At the same time, the analysis relies on the categories of US politics, particularly the distinction between Republicans and Democrats. This tension should be clarified as early as possible in the paper.

- We thank the reviewer for this remark, and have added a section to the introduction to clarify the geographic scope of our analysis. This section also addresses the apparent tension between finding two distinctly U.S. based clusters and the global nature of our data set. In particular, we note that since we did not limit data collection to the U.S. nor any other territory, the prevalence of American politics within our analysis is partly due to our focus on English-language discourse; partly due to the fact that the vast majority of Twitter’s English speaking user-base is located in the United States; and partly a reflection of the extent to which American political cleavages define global online discussions, at least when those discussions are carried out in English. Hence, even though we distinguish 'Democratic' and 'Republican' clusters of users, it is not the case that all users in our data set are based in the United States, for neither Public Health Institutions nor Antivaxxers are unique to U.S. discourse. 

Second, the link between vaccines and COVID was a bit confusing to me, given the time frame of the analysis (December 2019-June 2020). In that period, the emphasis was not on COVID vaccines, but more on the nature of the disease and mitigation strategies such as masks and containment. The authors do frame the study as an analysis of changes of the vaccination discourse following the emergence of COVID, but they could be more explicit that the discourse is on vaccines _in general_. Currently, readers may have the wrong expectation that the paper is about COVID vaccines.

- We thank the reviewer for this comment, it is important that the scope and purpose of the paper is clearly understood.

In order to clarify that the scope of the paper is how vaccine discourse in general changed due to Covid, we made modifications both to the Abstract and the Introduction, so that it becomes clear from the beginning.

Third, the pre-post comparisons are the most interesting parts of the analysis, but they could be discussed more in depth. For example, we see in Figure 1 that the structure of the retweet network changes significantly, but the point is barely elaborated in the text.

- There are two central points about the before and after networks that are emphasized in the text now. First, that even before the pandemic communities already exhibited some polarization. Second, that engagement increased substantially after the pandemic declaration. We develop the second point in more detail through the study, in Table 2, of the ratios and total numbers of retweet behavior.

Finally, Figures 5 and 7 are not legible, partly because of the low resolution but also due to the lack of a legend and the small size of the points. These figures should be improved.

- We now increased the resolution (dpi) of both figures, increased the font and point sizes in order to make those figures more legible. One problem is that image quality is greately reduced in the compilation of the revision document on this webpage. The images submitted have a greater dpi and are clearly legible. We do not know what else we can do in this regard.

---

## [Decision Letter · Decision Letter 2]

25 Oct 2022

Polarization and trust in the evolution of vaccine discourse on Twitter during COVID-19

PONE-D-21-40016R2

Dear Dr. Ojea Quintana,

We’re pleased to inform you that your manuscript has been judged scientifically suitable for publication and will be formally accepted for publication once it meets all outstanding technical requirements.

Kind regards,

Hossein Kermani

Academic Editor

PLOS ONE

Additional Editor Comments (optional):

**Congratulations!**

Reviewers' comments:

Reviewer's Responses to Questions

**Comments to the Author**

1. If the authors have adequately addressed your comments raised in a previous round of review and you feel that this manuscript is now acceptable for publication, you may indicate that here to bypass the “Comments to the Author” section, enter your conflict of interest statement in the “Confidential to Editor” section, and submit your "Accept" recommendation.

Reviewer #4: All comments have been addressed

Reviewer #7: All comments have been addressed

2. Is the manuscript technically sound, and do the data support the conclusions?

Reviewer #4: Yes

Reviewer #7: Yes

3. Has the statistical analysis been performed appropriately and rigorously? 

Reviewer #4: Yes

Reviewer #7: Yes

4. Have the authors made all data underlying the findings in their manuscript fully available?

Reviewer #4: Yes

Reviewer #7: Yes

5. Is the manuscript presented in an intelligible fashion and written in standard English?

Reviewer #4: Yes

Reviewer #7: Yes

6. Review Comments to the Author

Reviewer #4: All comments have been addressed by the authors.

Still some grammatical errors exits, check and resolve the mistakes.

Reviewer #7: The authors have addressed my comments satisfactorily. I have no other comments.

Some additional words to meet word count.

7. PLOS authors have the option to publish the peer review history of their article (what does this mean?). If published, this will include your full peer review and any attached files.

Reviewer #4: No

Reviewer #7: No

---

## [Editor Report · Acceptance letter]

21 Nov 2022

PONE-D-21-40016R2 

Polarization and trust in the evolution of vaccine discourse on Twitter during COVID-19 

Dear Dr. Ojea Quintana:

I'm pleased to inform you that your manuscript has been deemed suitable for publication in PLOS ONE. Congratulations! Your manuscript is now with our production department. 

Kind regards, 

on behalf of

Dr. Hossein Kermani 

Academic Editor

PLOS ONE